# An Offline Signature Verification and Forgery Detection Method Based on a Single Known Sample and an Explainable Deep Learning Approach

**Hsin-Hsiung Kao * and Che-Yen Wen**

Department of Forensic Science, Central Police University, Taoyuan 33304, Taiwan; cwen@mail.cpu.edu.tw

*** Correspondence: k@email.cib.gov.tw

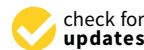

**Featured Application: This deep learning-based method is useful in the computer-aided detection of forgery signatures by using a single known genuine sample.**

**Abstract:** Signature verification is one of the biometric techniques frequently used for personal identification. In many commercial scenarios, such as bank check payment, the signature verification process is based on human examination of a single known sample. Although there is extensive research on automatic signature verification, yet few attempts have been made to perform the verification based on a single reference sample. In this paper, we propose an off-line handwritten signature verification method based on an explainable deep learning method (deep convolutional neural network, DCNN) and unique local feature extraction approach. We use the open-source dataset, Document Analysis and Recognition (ICDAR) 2011 SigComp, to train our system and verify a questioned signature as genuine or a forgery. All samples used in our testing process are collected from a new author whose signatures are not present in the training or other stages. From the experimental results, we get the accuracy between 94.37% and 99.96%, false rejection rate (FRR) between 5.88% and 0%, false acceptance rate (FAR) between 0.22% and 5.34% in our testing dataset.

**Keywords:** handwritten signature verification; signature examination; convolutional neural network (CNN); explainable deep learning; forensic science

## 1. Introduction

In forensic labs, handwriting examination is mainly performed by trained experts, and the result is primarily based on their experience and domain knowledge. For instance, in Taiwan, most handwritten signature examinations are performed by forensic document examiners of the Criminal Investigation Bureau (CIB). The examiners usually need to collect several signature samples from the suspect to capture handwriting features within available samples. However, it is hard to say how many signature samples would be "enough" for each case, which also depends on the expert's experience and ability.

It is typically not easy for police officers in first-line law enforcement to get enough reference signature samples for examination. In order to carry out the handwriting examination, forensic experts need to collect more signature samples from the suspect. Although more reference samples can promote reliability, they may not be available in all cases.

In practice, not all signature verification cases will reach the forensic examination stage. In some typical application scenarios such as industry, business, etc., we need to verify signatures in a quick way. For instance, investigators usually require a quick signature examination to evaluate the possibility or guide their investigation when the case is still under the preliminary investigation step. Another example is the use of commercial signature verification. When considering customer convenience, some companies even keep only a single signature for verification.

The automatic off-line signature verification solutions can be classified into two categories: handcrafted feature extraction algorithms and deep learning methods [1]. The deep learning methods are especially considered to be the most promising approach for its great capability for image recognition and detection. Although studies of deep learning with small-scale data are getting considerable attention in recent years, most of deep learning methods still need a large number of samples to train their system. In other words, most of the studies still need several (more than one) signature samples to accomplish their training process [1–4].

In this paper, we propose an off-line handwritten signature verification method using a single known genuine signature. The most challenging issue is how to train the network system with insufficient features of available samples. To conquer the challenge, we propose two alternative strategies to make our training feasible:

First, a small amount of samples does not necessarily mean a lack of features, and the number of features also depends on the extraction method. More specifically, there are several local features scattered throughout the entire signature, and these local features play an important role in signature verification. Therefore, we design a system based on an explainable deep learning method [5] and unique local feature extraction approach that focuses on verifying local signature blocks (rather than the whole signature image that commonly used in other research work).

Second, since we are dealing with a binary classification problem (genuine and forged), we can shift the focus of our system to learn "what is forged signature." We let our system learn a lot of features from forged signatures (of which it is easy for us to obtain a sufficient amount of samples) and make up for the disadvantage of insufficient genuine signature features.

The remainder of this paper is divided into five sections: Section 2 provides a brief discussion of automatic signature verification methods used in the paper. Section 3 introduces some key concepts and describes our methodologies. Section 4 explains the design of experiments and data collection. Section 5 shows the experiment results, evaluates the performance, and visualizes the CNN decision to make our model more interpretable. Finally, in Section 6, we conclude with the advantages and limitations of our approach and describe our future works.

## 2. Background and Related Work

### 2.1. Automatic Signature Verification

Automatic signature verification has been developed for decades. However, compared to other biometric approaches such as fingerprints, handwritten signatures have relatively high intra-class variability (a high variability between the signatures of a specific person) as well as low inter-class variability (skilled forgeries can be very similar to the genuine one). For these reasons, although signature verification has been widely used in the field of personal authentication, it is one of the most challenging issues in biometric technology, especially when we perform signature verification based on a single known sample. It is because we don't have enough samples to exclude the high-variability factor.

During this generation, deep learning has been one of the biggest breakthroughs in the artificial intelligence (AI) research area. Deep learning improves the deficiencies of traditional artificial neural networks by extending the depth, scale, and complexity of the network layers. This makes deep learning become a superb way to solve most kinds of pattern recognition problems, such as signature verification. Although deep learning-based signature verification methods have achieved great breakthrough in recent years, most of them still require several (more than one) genuine reference signature samples for training the networks [1–4].

Comprehensive surveys and state of the art reviews of the recent literatures can be found in the works of [1,4]. Among numerous papers, a growing number of researchers have realized that "learning with small datasets" will be a key factor for the success of signature verification in real practical scenarios. Authors such as Bouamra et al. [6] try to train the model with a small number of signature

samples through a one-class support vector machine (OC-SVM) classifier. Hafemann et al. [7] propose a solution based on a meta-learning approach.

### 2.2. Feature Extraction

Feature extraction is the key step for signature verification and can be generally classified into two types of methods: handcrafted feature extraction and feature learning methods [1]. In the handcrafted feature extraction methods, users design the extractor based on subjective perception. A number of survey papers [1,4,8–11] have reviewed the handcrafted feature extraction and classification methods used in signature verification. Deng et al. [12] used a wavelet-based feature extractor to calculate the curvature features of signatures. Pal et al. [13] took the local binary patterns (LBP) and uniform local binary patterns (ULBP) as their texture-based feature extraction method.

On the other hand, feature learning methods can extract features automatically without human intervention. This kind of method is commonly referred to as CNN and other deep learning approaches and has shown excellent performance compared to handcrafted features in many computer vision areas. Khalajzadeh et al. [14] proposed a deep CNN approach to learning features directly from signature image pixels for writer classification. Hafemann et al. [15] proposed a CNN-based method that can further learn stable features from variable-size signatures.

### 2.3. Signature Verification by Single Reference Sample

Signature verification based on only a single reference sample is a big challenge, because it is difficult to extract useful features from a sample of limited size. That is the reason that there has thus far been relatively little research in this area. Diaz et al. [16] proposed one of the state of the art practices nowadays of considering only one signature for training, but their work was focused on online (dynamic) signature verification. As for offline signature verification, Adamski and Saeed [17] used sampling techniques to acquire the feature vector from a one-pixel-wide signature. However, their study only dealt with random forgery samples (by using other author's genuine signatures as forgeries), rather than skilled forgeries (forgers imitate the genuine signature) that are used in our work.

## 3. Introduced Concepts and Methodologies

### 3.1. Convolutional Neural Network (CNN)

A convolutional neural network (CNN) [18] is a class of deep learning networks that has achieved state-of-the-art performance in many computer vision areas, such as image classification, pattern recognition, object detection, etc. Typically, CNN consists of three main types of components: convolutional layer, pooling layer, and fully-connected layer, as illustrated in Figure 1.

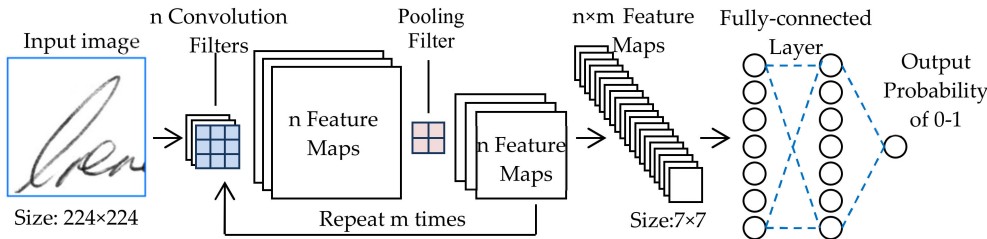

**Figure 1.** Schematic diagram of CNN, using the Vgg19 architecture for example [19].

The convolutional layer [20] uses multiple convolution filters (or convolution kernels) to extract the higher-level features from the low-level information, such as detecting the edges, corners, connection points, and other features from input images. Since multiple convolution filters will dramatically increase the size of the feature map and accompanied by tedious calculations, we use the pooling

layer [21] to reduce the feature map size and thus leads to a faster convergence rate for networks. Finally, all of the multi-dimensional feature maps are converted into a one-dimensional feature vector and input to the fully-connected layer. The fully-connected layer is basically a regular multilayer perceptron (MLP) and is used to generate class predictions for the further classification task. In this paper, we use the VGG19 architecture [19] and Inception V3 architecture [22] in our experiments, since both architectures are well-designed and have shown their great ability in ImageNet competition [23].

## 3.2. Explainable Deep Learning

Although deep learning models have shown their superior performance in various areas, they often lack interpretability. That makes them hard to adopt for forensics and other areas that require rigorous evidence. Therefore, explainable (or interpretable) deep learning methods have attracted more and more attention in recent years. In the field of computer vision, Simonyan et al. [5] proposed a gradient-based visual saliency method to visualize the decision-making process of neural networks. Its main idea is to show the image pixels (a saliency map) which is sensitive to the predictions of a network. This saliency map is obtained by computing the gradient of the class-specific score from a given classifier. The gradient indicates how much the change in a pixel that influences the classifier output. The gradient map itself can be considered as the saliency map in our cases, as shown in Figure 2.

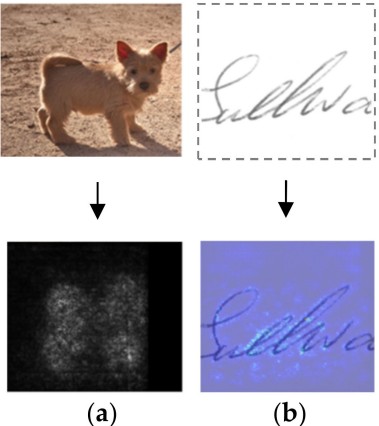

(a)        (b)

**Figure 2.** Saliency maps for the predicted class. (**a**) Dog and its saliency map marked by grayscale (taken from [5]); (**b**) signature and its saliency map marked by jet-colormap (taken from our experiment).

## 3.3. Design Principles

Although deep learning-based methods have demonstrated their great capability for signature verification, most of these classifiers still rely on large-scale (or at least a certain large size) datasets to process learning tasks. Undoubtedly, it will reduce the applicability of the signature verification system, therefore, the primary objective of this paper is to address this challenge with only single reference sample. Furthermore, another objective is to make our verification results more acceptable by visualizing CNN networks' decisions and providing appropriate explanations.

This paper examines assumptions that our method can be used even under the extreme condition of only single genuine reference sample, by extracting sufficient and useful local features from additional forgeries. Consequently, we designed 12 experiments with different sample sizes to confirm our assumptions and further enhance its effectiveness by using visual interpretation techniques.

The final verification results can be obtained by the voting method with different threshold and can be adjusted according to different types of application requirements. For example, in courts of law, this threshold will be chosen to be very high, whereas for typical business applications, a balance must be sought between verification accuracy and customer convenience.

### 3.4. Strategies for Using Only Single Genuine Reference Signature

As mentioned in Section 2.1, handwriting signatures have high intra-class variability. This key factor presents a particularly difficult condition for signature verification with single reference signature. Some studies can overcome this disadvantage by adding additional signature samples for training or by other feather extraction methods [1–4]. In this paper, we deal with the database with a constrained sample size. Therefore, we propose the following two main strategies to solve our problem of insufficient samples:

1.  We assume that there are a lot of useful local features (such as the features of strokes lines, starting, turning, and ending points) generally scattered across the whole signature image. Based on this condition, we convert a single signature image into many overlapping sub-image blocks. Since our method has the capability of extracting local features, these sub-image blocks can be used as training samples. In addition, we also apply some data preprocessing techniques which are commonly used for deep learning and image processing tasks (such as sample screening and data augmentation). In this way, the number of training samples can be expanded. Further details will be explained in Section 3.5.
2.  Since we are dealing with a binary classification problem, we can define what is a genuine signature when our trained system can successfully detect the forged signatures. That is, we let the network learn more features from forged signatures.

Fortunately, samples of forged signatures are relatively easy to obtain and basically unlimited in number (because it can be generated by artificial imitation or computer algorithms). In a later section we will use experimental results to show that our trained network is good at learning some common forgery features which exist in different signatures and authors. As more forged samples are added, the accuracy of the network will be higher.

### 3.5. Data Preprocessing

Before extracting the features by our CNN network, some data preprocessing must be performed: background noise reduction, data augmentation, and screening. The goal of our preprocessing approach is to increase the number of samples by converting a single original signature image into many sub-image blocks, and to filter the sub-image with enough stroke pixels as the training dataset. For example, the No.1 genuine signature from author ID 14 is converted to 6.828 new sub-images for the VGG19 training dataset (see Figure 3). The steps of the preprocessing are shown as follows:

1.  We convert the raw image into a grayscale image and then save it as a 24-bit BMP file (as the first step in Figure 3).
2.  We use a block-based method for data augmentation as the second step in Figure 3. We set a window as a sliding mask to get a sub-image block from the original image. The window size depends on the network architecture. In our case, the VGG19 network uses $224 \times 224$ window size and the Inception-v3 network uses $299 \times 229$ one. Then we shift the mask window by 20 pixels each time to repeat the process from left-to-right then top-to-bottom. After finishing the processes above, we can obtain about 6.828 overlapping sub-image blocks from an original signature image.
3.  To reduce paper texture and background noise caused by the optical scanning unit, we brighten the sub-image by 7.5%, and the method is to multiply each RGB pixel values by 1.075 directly. In our experiment, the background noise is biased towards light colors. Therefore, this method can remove most of the noise without destroying the features of the handwriting strokes (the excessive brightness may damage the features of handwriting signature).
4.  We rotate each sub-image blocks clockwise by a predefined angle and repeat the process until a full 360 degrees rotation is done. Note that our experiment involves 6 sub-datasets includes: genuine signature for training, forged signature for training, genuine signature for verification, forged signature for verification, genuine signature for testing, and forged signature for testing.

For each sub-datasets, we set the rotation angle as follows: the genuine signature used for training is rotated by 10 degrees each time. The forged signature used for training is rotated by 20 degrees each time, and the rest of the sub-datasets are only used for verification and testing, so we set it to 60 degrees to save the computation cost, see Table 1. In order to prevent the large class-skew of our training dataset (due to the number of genuine samples is far less than the forgeries), as shown in Figure 4a. We deliberately use a smaller rotation angle (10 degrees) to increase the sample size of the genuine signature dataset which is used for training.

5.  We check each sub-image block to see if it contains enough information for feature extraction and set some inspection thresholds based on experience. We set 250 as the threshold grayscale value, and consider a pixel whose grayscale exceeds the threshold as a valid pixel. Then, if a sub-image block has over 7.5% valid pixels, we classify it as a valid sample. Conversely, we regard it as an invalid sample and drop it.

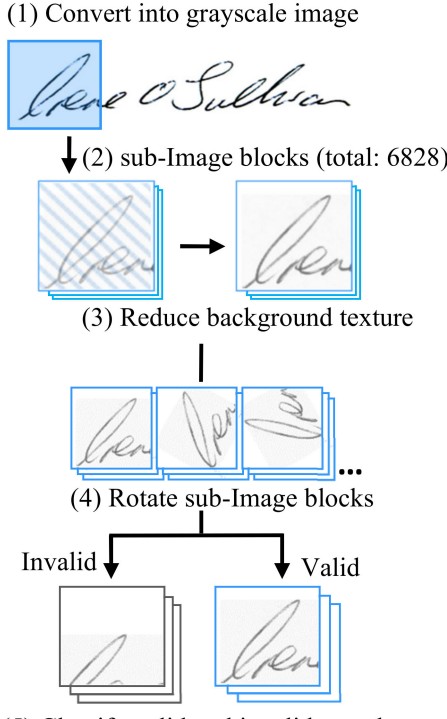

(1) Convert into grayscale image

(2) sub-Image blocks (total: 6828)

(3) Reduce background texture

(4) Rotate sub-Image blocks

Invalid      Valid

(5) Classify valid and invalid samples

**Figure 3.** Preprocessing to increase the sample size.

**Table 1.** The rotation angle for data preprocessing.

|  | Training | Validation | Testing |
|---|---|---|---|
| Genuine | 10 |  |  |
| Forged | 20 | 60 | |

By performing the above preprocessing steps, these sub-image blocks are theoretically qualified as training samples. Furthermore, our preprocessing methods not only augment the number of samples and increase the feasibility of deep learning, but also have the following benefits:

Firstly, using sub-image blocks as the training and verification data can effectively prevent a small number of local features from dominating the whole CNN system. Secondly, by applying the rotation process, we can make our CNN system focus on the rotation-invariant features and reduce the unnecessary influence from different handwriting angles. Thirdly, the created rotation sub-image

blocks can simulate the signature intra-class variability to a certain extent. And we have found that increasing the number of rotations can help improve system performance.

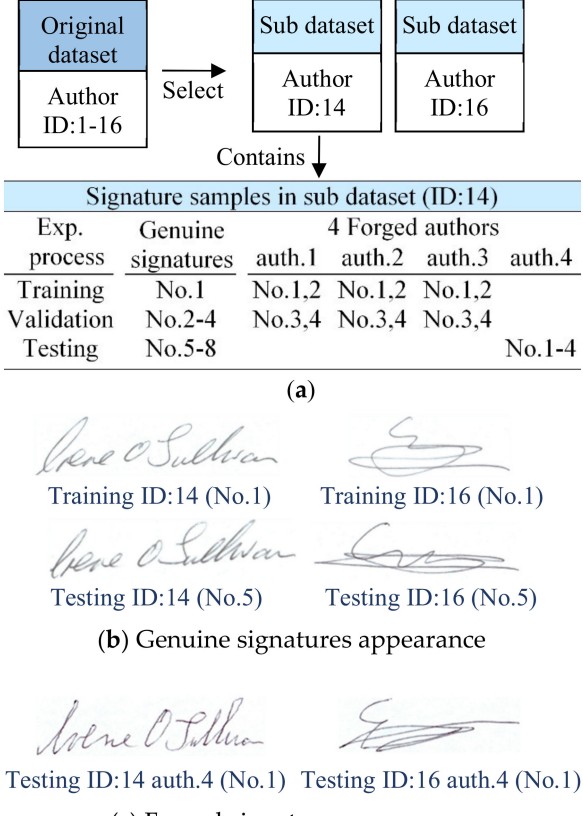

(a)

Training ID:14 (No.1)　　　　Training ID:16 (No.1)

Testing ID:14 (No.5)　　　　Testing ID:16 (No.5)

(**b**) Genuine signatures appearance

Testing ID:14 auth.4 (No.1)　Testing ID:16 auth.4 (No.1)

(**c**) Forged signatures appearance

**Figure 4.** (**a**) Sub-datasets used for experiment; (**b**) genuine signatures appearance in training stage; (**c**) forged signatures appearance in testing stage.

## 4. Experiments

We use open-source resources in our experiments. Our signature samples are from ICDAR 2011 SigComp dataset [24]. We use Python Imaging Library Pillow 5.0 [25] for data preprocessing. Finally, we train and evaluate our network by using deep learning frameworks: TensorFlow 1.7 [26] and Keras 2.1 [27]. All experimental processes were run on a PC with an Intel Core 4 GHz CPU (eight cores, 8 M cache) and 12 GB memory.

### 4.1. Data Collection

CDAR 2011 SigComp dataset is used for signature verification competition and contains both online and offline signature samples. The offline section of this dataset has different sample sizes of skilled forgeries for each genuine author, see Table 2.

In Table 2, we can see there are very limited forgery samples in the dataset. However, author ID No. 14 and No. 16 contain the largest number of forged reference signatures (each author has 16 skilled forgeries from four authors). In this paper, we need a certain number of forged signature samples, which are not only used for training the network, but also for further evaluating the performance. Therefore, we use the No. 14 and No. 16 author's signatures as experimental samples. That is, our experimental database includes two genuine authors (each genuine author has eight signature samples) and eight forgers (each forger has four signature samples). Note that in the original SigComp dataset, the number of genuine signature samples far exceeds the needs of our experiment (there are 24 signature samples for each genuine author), so we just take the first eight samples based on the ascending order of signature's ID number. For the simplicity sake, we denote them as No. 1–No. 8.

No. 1 as the training data, Nos. 2−4 as the validation data, and Nos. 5−8 as the test data, as shown in Figure 4a.

**Table 2.** The summary of the forged signature samples in SigComp 2011.

| Genuine Authors ID | Forgeries Samples | |
|---|---|---|
| | Author Num | Signature Num |
| 001 | 2 | 8 |
| 002 | 3 | 12 |
| 003 | 3 | 12 |
| 004 | 3 | 12 |
| 006 | 3 | 12 |
| 009 | 3 | 12 |
| 012 | 3 | 12 |
| **014** | **4** | **16** |
| 015 | 3 | 12 |
| **016** | **4** | **16** |

http://www.iapr-tc11.org/mediawiki/index.php/ICDAR_2011_Signature_Verification_Competition_ (SigComp2011).

It must be noted that samples of author ID: 16 have the high intra-class variability. The genuine signatures used for training and testing processes have very different styles, as shown in "ID: 16 (No. 1)" and "ID: 16 (No. 5)" in Figure 4b. Obviously, it is difficult to verify if these signatures are from the same person, even with human visual comparison. This will also affect the performance of the network. We will discuss later.

In Figure 4a, each sub-dataset contains eight genuine signatures from the single genuine author, and 16 forged signatures from four different authors. We use the last author (No. 4) for performance evaluation (not used in the training process). The appearances of the signatures are shown in Figures 4b and 3c.

## 4.2. Experimental Design

We designed 12 experiments with two different network architectures, two genuine authors, and three types of forgery sample size, as shown in Table 3.

**Table 3.** The summary of the datasets used in 12 experiments.

| Network | Genuine Author | Exp. ID | Stage | Genuine Signature | Forgery Author | | | |
|---|---|---|---|---|---|---|---|---|
| | | | | | Auth. 1 | Auth. 2 | Auth. 3 | Auth. 4 |
| VGG19/ **Inception V3** | ID:14 | Exp. 1/ **Exp. 7** | Training | No. 1 | No. 1,2 | No. 1,2 | No. 1,2 | |
| | | | Validation | No. 2−4 | No. 3,4 | No. 3,4 | No. 3,4 | |
| | | | Testing | No. 5−8 | | | | No. 1−4 |
| | | Exp. 2/ **Exp. 8** | Training | No. 1 | No. 1,2 | No.1,2 | | |
| | | | Validation | No. 2−4 | No. 3,4 | No.3,4 | | |
| | | | Testing | No. 5−8 | | | | No. 1−4 |
| | | Exp. 3/ **Exp. 9** | Training | No. 1 | No. 1,2 | | | |
| | | | Validation | No. 2−4 | No. 3,4 | | | |
| | | | Testing | No. 5−8 | | | | No. 1−4 |
| | ID:16 | Exp. 4/ **Exp. 10** | Training | No. 1 | No. 1,2 | No. 1,2 | No. 1,2 | |
| | | | Validation | No. 2−4 | No. 3,4 | No. 3,4 | No. 3,4 | |
| | | | Testing | No. 5−8 | | | | No. 1−4 |
| | | Exp. 5/ **Exp. 11** | Training | No. 1 | No. 1,2 | No. 1,2 | | |
| | | | Validation | No. 2−4 | No. 3,4 | No. 3,4 | | |
| | | | Testing | No. 5−8 | | | | No. 1−4 |
| | | Exp. 6/ **Exp. 12** | Training | No.1 | No. 1,2 | | | |
| | | | Validation | No. 2−4 | No. 3,4 | | | |
| | | | Testing | No. 5−8 | | | | No. 1−4 |

We arrange forgery samples of different sizes as controlled experiments. It is used to check whether our networks can learn some common and ubiquitous forgery features (exist within signatures of different forgers) to discriminate between genuine signatures and forgeries. Meanwhile, in order to evaluate the impact of different network performance on the experiment results, we use two different architecture networks (VGG19 / Inception V3) in our experiments.

Please be noted that we intentionally arranged two network architectures with different performance for comparison experiments. Inception V3 has better performance than VGG19 in almost all aspects. This is because the former adopts the deeper network. In general, then depth of layer leads to better accuracy [19]. Thence, in 12 experiments, Exp. 7 and Exp. 10 use the higher-performance Inception V3 architecture with the most complete dataset as the main experiment to evaluate systems performance, and the remaining experiments are served as the control group.

### 4.3. Transfer Learning and Per-Training

Transfer learning [28] is a kind of machine learning technique. It takes the existing model weights from a solved problem which is different but similar to ours. By transferring the well-trained weights to our new network model can dramatically improve performance with less training time. Therefore, we use the trained weights from ImageNet competition for VGG-19 and Inception V3 architecture for our models. These two networks are built to recognize 1000 different categories from target images (such as strawberry, bike, balloon, etc.), and have already achieved great success in ImageNet competition. We use those weights as the initial weights of our models, for they are already finely tuned and do exceptionally well in object classification. This allows us to train our network more effectively.

After applying the initial weights, we still have to fine-tune the network for the better accuracy. In this step, we only have to per-train the last three layers of the networks for just one epoch (freeze all previous layers). This prevents the early training process from undermining valuable initial weights.

### 4.4. Networks Training and Validation

In the training process, we use a loss function (or cost function) to estimate the deviation between the predicted value and the actual label, and then train the network to minimize the loss value. The lower the loss value, the closer the predicted result is to the real label. Our output layer is a sigmoid function that handles the binary problems and generates an S-shaped curve whose values between 0–1, as shown in Equation (1). We choose a cost function called binary cross-entropy (BCE) as shown in Equation (2), where $\hat{y}$ represents the predicted probability in which target is a genuine signature and y represents the correct one [29]:

$$\text{Sigmoid}(x) = \frac{1}{1 + e^{-x}} \tag{1}$$

$$\text{BCE} = -y \, \log\!\left(\hat{y}\right) - (1 - y) \log\!\left(1 - \hat{y}\right) \tag{2}$$

Then we use a popular optimization technique called the stochastic gradient descent (SGD) algorithm [30] to optimize our neural networks by minimizing the BSE value in Equation (4). Since an extortionate learning rate may hinder the convergence, we set a relatively small value of $e^{-4}$ and set the momentum parameter as 0.9, which is most often used in SGD. Our dataset includes 48 images with two genuine and eight forged authors. The sample allocation is according to the ascending order of signature ID number, as refer to Table 3. All these images are converted into 345,874 sub-image blocks for the VGG19 network and 213,449 sub-image blocks for the Inception-V3 network experiment. After per-train the networks for one epoch, our main training process is finished in two epochs for the author No. 14 dataset, and six epochs for the author No. 16 dataset. The training epochs are different due to author ID: 16 signatures have higher intra-class variability therefore the corresponding network needs more epochs to converge properly.

From Exp. 1 to Exp. 12, we get nearly 100% training and validation accuracy. It indicates that our networks sufficiently capture the features from training set and are able to correctly distinguish between

the genuine signatures and the forgeries. However, although the above training and verification stages are based on different signatures, they still come from the same authors. To further verify whether the networks can be used by completely unknown authors, we still need to run the testing stage (which will be explained in the next section).

## 5. Results and Discussion

### 5.1. Networks Performance

After the network training and validation, to further test whether the networks can be used by completely unknown author, we test the networks with new authors' signatures that are not present in the training and verification dataset stage (see Table 3). We then used the following metrics to evaluate the performance of our network: training, validation and testing accuracy; false rejection rate (FRR) or type I error ($\alpha$ error); false acceptance rate (FAR) or type II error ($\beta$ error), which are defined in Equations (3)–(5) [31]:

$$\text{Accuracy} = \frac{\text{TP} + \text{TN}}{\text{TP} + \text{FP} + \text{FN} + \text{TN}} \times 100\% \tag{3}$$

$$\text{FRR} (\alpha) = \frac{\text{FN}}{\text{TP} + \text{FN}} \times 100\% \tag{4}$$

$$\text{FAR} (\beta) = \frac{\text{FP}}{\text{FP} + \text{TN}} \times 100\% \tag{5}$$

TP is the number of true positive, which indicates the genuine signatures are correctly classified as positive. FP is the number of false positive, which means the forged signatures are misdetected as positive. Likewise, we got false negative (FN) and true negative (TN). As a result, the higher accuracy and the lower error rate (FRR/FAR) can be considered as better performance. In our experiments, different results obtained from different training sample sizes (numbers of forgery authors and signatures for training) and network architecture, which are summarized in Table 4.

**Table 4.** The summary of experimental results.

| Network Architecture | VGG-19 | | | | | | Inception V3 | | | | | |
|---|---|---|---|---|---|---|---|---|---|---|---|---|
| **Genuine Author ID** | ID:14 | | | ID:16 | | | ID:14 | | | ID:16 | | |
| **Experiment No.** | Exp. 1 | Exp. 2 | Exp. 3 | Exp. 4 | Exp. 5 | Exp. 6 | Exp. 7 | Exp. 8 | Exp. 9 | Exp. 10 | Exp. 11 | Exp. 12 |
| Forged author Num. | 3 | 2 | 1 | 3 | 2 | 1 | 3 | 2 | 1 | 3 | 2 | 1 |
| Forged signature Num. | 6 | 4 | 2 | 6 | 4 | 2 | 6 | 4 | 2 | 6 | 4 | 2 |
| Training accuracy (%) | 99.78 | 100 | 100 | 100 | 99.88 | 99.98 | **99.93** | 100 | 99.67 | **100** | 99.97 | 100 |
| Validation accuracy (%) | 99.42 | 97.66 | 100 | 89.97 | 94.26 | 84.58 | **100** | 100 | 95.66 | **98.96** | 97.56 | 99.98 |
| Testing accuracy (%) | 97.75 | 97.10 | 96.70 | 53.77 | 56.86 | 48.38 | **99.96** | 99.98 | 76.93 | **94.37** | 90.23 | 90.85 |
| Test FRR (%) | 2.95 | 3.52 | 0.4 | 45.19 | 40.98 | 73.43 | **0** | 0 | 27.81 | **5.88** | 15.16 | 2.83 |
| Test FAR (%) | 1.08 | 1.89 | 8.06 | 47.39 | 45.55 | 27.27 | **0.22** | 0.07 | 1.47 | **5.34** | 3.66 | 16.31 |

### 5.2. Visualize the CNN Decision

In order to make our verification results more explainable. We generate an attention saliency map over the questioned sub-image block, which is based on the last dense layer (fully-connected layer) outputs in our networks. For clearer visualization, we use the jet-colormap to represent the intensity then impose the saliency map on the original signature image, see Figure 5a. The jet-colormap returns the colors contain blue, green, yellow, and red. It represents the values of saliency (between 0–1), the color scheme as shown in Figure 5b.

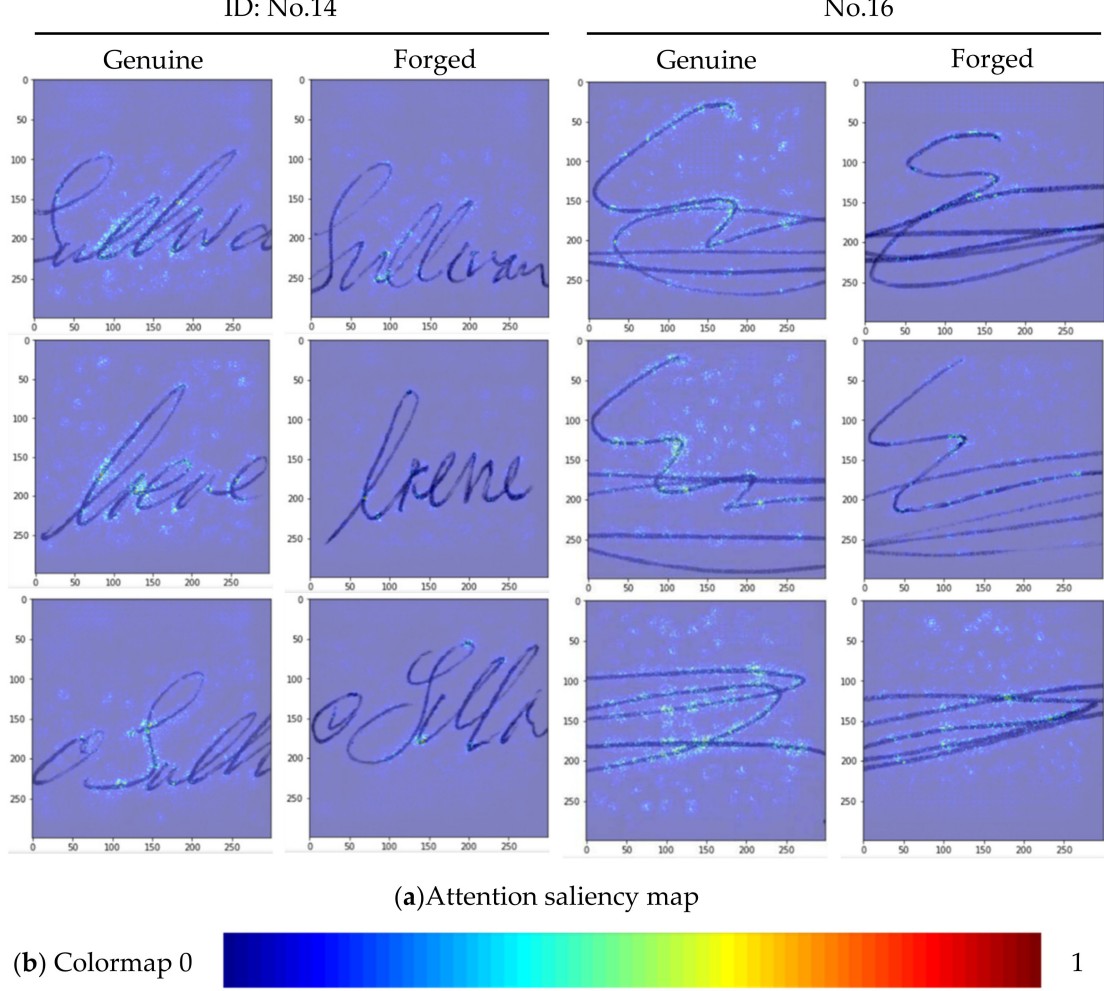

**(a)**Attention saliency map

**(b)** Colormap 0 [====colormap bar====] 1

**Figure 5.** Saliency maps for the output layer of CNN networks. (**a**) Attention saliency maps for the signature sub-image blocks. (**b**) The colormap contains values between 0 and 1.

In Figure 5, the images are from authors No. 14 and No. 16. The network architecture is the higher-performance Inception V3 network and trained by a genuine author and three forged authors (i.e., the main experimental group: Exp. 7 and Exp. 10). These attention saliency maps are able to provide some insights into our deep CNN network. First of all, we can confirm that the verification results of our network are derived from the signature itself (rather than other unrelated noise). Furthermore, by observing the significant areas, it can show which strokes are important for the networks decision (i.e., the highly saliency areas). For example, we find that the turning point and the intersection of the strokes are often used as an important basis for signature verification.

Compared to the saliency maps of other general image classification networks, our saliency maps rarely show high saliency areas (marked in red). This is in line with the experimental design of this paper because our networks use the only single known genuine signature for training. Considering that each individual author's signatures have high intra-class variability, in general, we need a certain number of signatures to extract the stable and deterministic features from them. Therefore, networks trained by a single signature are "not" enough to extract this kind of feature. Also, our work focuses on verifying by using local features with signature's sub-image blocks. On the other hand, we want to avoid such extreme features from dominating the whole network decision.

*5.3. Discussion*

The experiment results show the capability of the proposed method, especially for our main task (Exp. 7 and Exp. 10). Our approach not only correctly distinguishes the genuine and forgeries in the training and validation period but also achieved 99.96% and 94.37% accuracy in Exp. 7 and Exp. 10 testing dataset, respectively. Please be noted that in the Exp. 10 (which uses the ID:16 dataset), it is very difficult to verify if these signatures are from the same author, even through human visual comparison (due to the high intra-class variability and low inter-class variability), as shown in ID: 16 of Figure 4b,c also presented on the right side of Figure 5a.

Since our training method is based on only a single known genuine sample, it may lead to poor performance, especially if the training data have relatively high intra-class variability. Thus, taking into account the fact that the experiment is carried out under such adverse conditions, the lower accuracy of Exp.10 is in line with expectations. The same reason can also explain why the attention saliency maps of ID No.16 in Figure 5a have more saliency areas than the background (compared with ID No.14). This is also due to the inevitable over-fitting under unfavorable training conditions. Furthermore, we can ensure the reliability of experimental results through the following three experimental designs.

First, we divide the experimental dataset into three parts for training, validation, and testing. Each signature sample only belongs to one part. Moreover, the signatures used in the testing stage are collected from a whole new author whose samples are not present in the training and validation processes. Secondly, the experimental results of the controlled group are also very useful. In Table 4, we can observe that the accuracy drops significantly with the decreasing number of forgery samples. Also, the accuracy decreases when we use a relatively poor-performing network architecture (Vgg19). It shows that the way we use to improve the accuracy rate is reasonable. Thirdly, we visualize the CNN networks' decision. We can observe that the experimental results of our network mainly come from the signature strokes rather than other unrelated features (e.g., paper background texture).

Finally, in order to explain the effectiveness of the proposed method, and also check whether it can be applied to different language type. We have further evaluated the model using real collection of Chinese signatures. Those signatures are collected from a genuine author and 4 forgers, and each forger contributes 4 signature samples by imitating the genuine one. All signatures are collected on A4 size white paper using the same ballpoint pen (0.1 mm black ink) and scanned by Canon UFR II scanner with 600 dpi resolution in 8-bit grayscale images. The rest of the process is basically the same as the main experiment (Exp. 7 and Exp. 10) of this study, and the result is also promising. We get the accuracy of 89.5%, FAR of 24.77% and FRR of 2.91% in this real collected Chinese dataset. Similarly, the saliency map in this case indicates that the model can effectively distinguish between genuine and forged strokes, as shown in Figure 6.

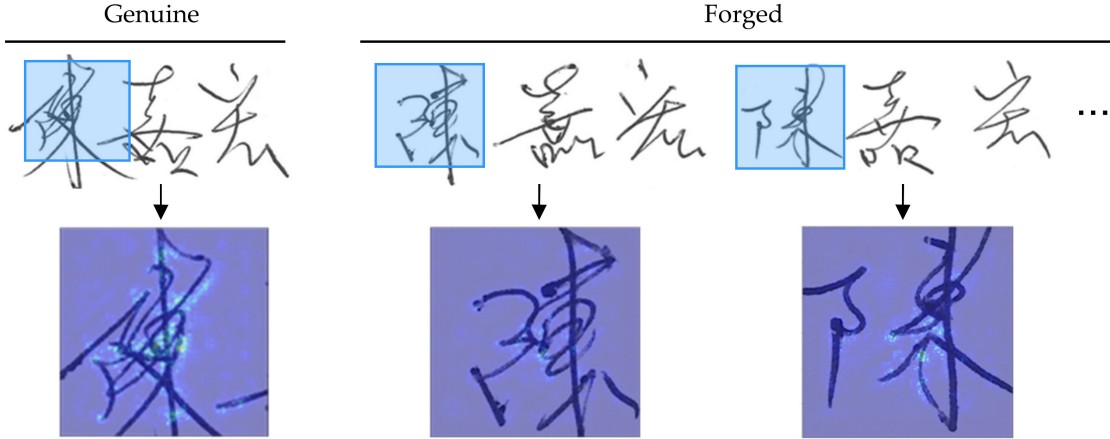

**Figure 6.** Saliency maps for real collected Chinese signatures.

However, such accuracy is still lower than our main experiments. We speculate that this is due to the complex structure of Chinese characters and accompanied by the higher intra-class variability. It may cause adverse conditions for CNN networks, especially when using only single reference signature.

## 6. Conclusions and Future Work

In this paper, we propose an off-line handwritten signature verification method by using single known sample and based on a deep CNN network. We ensure the reliability of the experimental results through a series of methods, including preprocessing (removing background noise), designing controlled groups for different sample sizes and network architectures, and applying visualization techniques (to provide interpretability of the model). The experimental results indicate that it is possible to perform automatic signature verification by single known sample. And our method can learn some useful features to discriminate among different signatures and authors. Even under the unfavorable conditions of small sample size, there is still a relatively high accuracy rate between 89.5% and 99.96%.

We enhance the effectiveness of our method by using visual interpretation techniques, and find the results are well consistent with human cognition. In addition, we find that the results of the controlled group are particularly instructive. The results show that even with the single known sample, the performance of the network can be effectively improved by increasing the number of forged samples. In practical applications, since more forgery samples will lead to a higher accuracy rate, we can create forged signatures by ourselves to further improve the performance of our proposed method.

Our method is currently used to solve binary classification problems. The output layer is a sigmoid function and output a p-value between 0 and 1. In this case, p represents the probability that the sub-image block comes from the genuine signature, so the probability of the other category is just 1-p. If we want to verify the final result, we can use the voting method based on the number of sub-images. However, if we need to further optimize the system performance and provide a convincing verification result, more data and follow-up studies are required. Our further work will focus on these subjects.

**Author Contributions:** Conceptualization, H.-H.K.; Methodology, H.-H.K. and C.-Y.W.; Software, H.-H.K.; Validation, H.-H.K. and C.-Y.W.; Writing—original draft, H.-H.K.; Writing—review and editing, C.-Y.W.; visualization, H.-H.K. All authors have read and agreed to the published version of the manuscript.

**Funding:** This research received no external funding.

**Conflicts of Interest:** The authors declare no conflict of interest.

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
