# Peer review of "An Offline Signature Verification and Forgery Detection Method Based on a Single Known Sample and an Explainable Deep Learning Approach"

_applsci, doi:10.3390/app10113716_

Round 1

Reviewer 1 Report

The authors proposed an approach for the Offline Signature Verification, based on a single known sample. This work presents a specific data augmentation method aiming to overcome the issue of a small amount of samples.

Overall, the authors well explained the problem and the proposed approach.  The idea of using extensive augmentation techniques to overcome the issue of a small amount of samples in deep learning is not new. However, this specific augmentation method would have some merit if it can achieve the high performance of Offline Signature Verification. The authors used a very small dataset, I suggest the authors  apply their method in other datasets such as GPDSsynthetic-Offline, GPDS-75, MCYT-75, UTSig, and CEDAR. By doing this, they can prove the effectiveness of the method.

Reviewer 2 Report

The submission addresses the challenge of distinguishing forgeries from legitimate hand-made offline signatures, which entail a problem widely studied and addressed by the research community. In this context, CovNets represent a common technological enabler habitual in the bibliography. The contributions attempt to go beyond the state of the art by assuming only a single known genuine signature as input of the modelling and learning algorithms developed. This differentiating aspect should be enough to justify  the novelty of the contribution for two reasons: 1) the problem is very challenging from the perspective  of deep learning, which typically relies on large datasets; and 2) the user acceptance is a critical indicator of the transferability of related solutions, where requests of large amounts of signatures may jeopardize the real applicability of the proposal. Because of this, this reviewer consider the submission solid enough for be considered as publication, but the following comments should be addressed:

  • There are a lot of recent and larger datasets than that considered for validation and benchmarking purpose. This decision should be more clearly justified; or in the opposite, the experimentation should be complemented by a more real collection.
  • The submission lack of a proper description of the design principles of the conducted research. A new section/subsection should be included highlighting: primarily/secondary objectives, null/alternative hypothesis, assumptions, application requirements, limitations, etc..
  • The introduced concepts and methodologies should be separated from the state of the art and experiments sections. For example, a new section should concentrate the description of the proposed algorithms
  • Overall, the state of the art is out of date. A subset of more recent publications should be included

Reviewer 3 Report

I recommend revising your paper based on the following comments:

Abstract

First, specify the FRR and FAR metrics what exactly they stand for, then their abbreviation.

Introduction

The first page, Introduction needs a reference …

Line 38-39: Change the wording of this sentence …

Background and Related Work

Line 81-82: Remove the article ‘the’ before artificial intelligence (AI) …

Line 85: Closing punctuation is missing …

Round 2

Reviewer 1 Report

Thank you for your revision version. I still do have concerns about the number of "genuine authors" in your experiments. Could you please elaborate why testing on the signatures of only 1 or 2 genuine authors is enough to prove the effectiveness and the generalization of your method? If the ICDAR_2011 dataset does not have enough forged signature samples, other datasets (such as MCYT_Signature, GPDS-960) can provide you with this condiditon.

Round 3

Reviewer 1 Report

OK. Please add your arguments (the explainability of signature verification) into the paper.